# A High-Throughput Microtiter Plate Screening Assay to Quantify and Differentiate Species in Dual-Species Biofilms

**DOI:** 10.3390/microorganisms11092244

**Published:** 2023-09-06

**Authors:** Víctor Campo-Pérez, Júlia Alcàcer-Almansa, Esther Julián, Eduard Torrents

**Affiliations:** 1Bacterial Infections and Antimicrobial Therapies Group, Institute for Bioengineering of Catalonia (IBEC), The Barcelona Institute of Science and Technology (BIST), Baldiri Reixac 15-21, 08028 Barcelona, Spain; vcampo@ibecbarcelona.eu (V.C.-P.); jalcacer@ibecbarcelona.eu (J.A.-A.); 2Departament de Genètica i de Microbiologia, Facultat de Biociències, Universitat Autònoma de Barcelona, 08193 Barcelona, Spain; 3Microbiology Section, Department of Genetics, Microbiology and Statistics, Faculty of Biology, University of Barcelona, 643 Diagonal Ave., 08028 Barcelona, Spain

**Keywords:** crystal violet, biomass quantification, *Pseudomonas aeruginosa*, *Burkholderia cenocepacia*, dual-species biofilms

## Abstract

Pathogenic bacteria form biofilms during infection, and polymicrobial biofilms are the most frequent manifestation. Biofilm attachment, maturation, and/or antibiotic sensitivity are mainly evaluated with microtiter plate assays, in which bacteria are stained to enable the quantification of the biomass by optical absorbance or fluorescence emission. However, using these methods to distinguish different species in dual-species or polymicrobial biofilms is currently impossible. Colony-forming unit counts from homogenized dual-species biofilms on selective agar medium allow species differentiation but are time-consuming for a high-throughput screening. Thus, reliable, feasible, and fast methods are urgently needed to study the behavior of polymicrobial and dual-species communities. This study shows that *Pseudomonas aeruginosa* and *Burkholderia cenocepacia* strains expressing specific fluorescent or bioluminescent proteins permit the more efficient study of dual-species biofilms compared to other methods that rely on measuring the total biomass. Combining fluorescence and bioluminescence measurements allows an independent analysis of the different microbial species within the biofilm, indicating the degree of presence of each one over time during a dual-species biofilm growth. The quantitative strategies developed in this work are reproducible and recommended for dual-species biofilm studies with high-throughput microtiter plate approaches using strains that can constitutively express fluorescent or bioluminescent proteins.

## 1. Introduction

Polymicrobial infections denote diseases involving multiple species of microorganisms, a situation commonly associated with chronic infections [1]. The interactions between pathogens can vary from competition for the same space or nutrients to cooperative interspecies mechanisms that enhance joint survival and the evasion of host responses and increase antibiotic susceptibility [2]. These infections have become more frequent and relevant in recent years and are especially worrisome because biofilm-forming microbes are associated with higher mortality rates [3]. Biofilms are composed of bacteria embedded in an extracellular polymeric matrix consisting of polysaccharides, protein, and extracellular DNA [4]. This growth behavior confers a broad range of advantages on pathogens, such as quorum sensing interactions and interspecies DNA exchanges that lead to metabolic cooperation and, in some cases, a high tolerance to antibiotics. This provides a competitive advantage and significantly enhances the resistance capacity inside the host [5]. Pathogenic biofilm-forming bacteria present an increasing multidrug resistance to the currently available antibiotics, so biofilm development is a relevant treatment target in the fight against chronic infections, and new anti-biofilm strategies are needed [6,7]. In this sense, over the last few years, strategies based on the delivery of anti-biofilm compounds, nanoparticles, and enzymes have been developed. Some studies have focused on inhibiting bacterial adhesion using compounds and enzymes that affect structures and biomolecules involved in cell adhesion, a key feature in the development of biofilms [8]. Others develop microparticles based on polymeric carriers, dispersibility enhancer molecules, and anti-biofilm compounds [9]. The use of metal oxide nanoparticles has reported significant anti-biofilm efficiencies, but cytotoxicity issues in eukaryotic cells makes this nanoparticle applicability controversial [7].

The different in vitro methods developed for biofilm growth studies are categorized into static, continuous flow or open systems, and microfluidic biofilm culture [10]. The microtiter plate assay is the most widespread system used to study the growth of static biofilms and can be used for high-throughput screenings. This consists of growing biofilm-forming microorganisms in polystyrene microtiter plates using a suitable culture medium. Cells become attached to the walls of the wells, usually at the culture medium–air interface, and start to develop a biofilm growth form. After washing the wells to remove the planktonic cells, the adhered cells are allowed [11] to develop a mature biofilm. Biofilm development is then analyzed by the indirect quantification of the biomass, with crystal violet cell staining as the most commonly used method [12]. This method presents several advantages, such as low economic cost, technical simplicity, easy setup, and applicability to the high-throughput screening of biofilm-forming strains [13], and represents an ideal proposal for an anti-biofilm drug testing platform [14]. However, crystal violet staining also causes concern due to its toxicity to the environment and carcinogenic effects [15,16]. In recent years, alternatives to crystal violet, such as classic safranin, another cationic dye that binds to negatively charged bacterial cell walls, and the genetic material fluorescent dye SYTO 9 or acridine orange [17], have been used. These two new dyes stain live and dead cells and extracellular matrix components, thus generally overestimating biomass values [18]. Nevertheless, both are used for optical absorbance or fluorescence measurements of monomicrobial [19] and dual-species biofilms [20]. An outstanding alternative solution to this problem is resazurin-based viability staining approaches [15,17,21], which allow the quantification of biofilm-forming live cells exclusively by the metabolic reduction of nonfluorescent resazurin to fluorescent resorufin. In addition, this method enables a correlation of the fluorescence emitted with the number of cells in the biofilm, although it must first be optimized and calculated for each species [21].

However, general biomass staining methods and resazurin assays, as described previously, are very limiting in studies of dual-species biofilms, since they stain nonspecifically, impeding the independent quantification of each individual microbial species present in the mixed biofilm. This is a recurring problem in these types of studies, which end up without the evaluation of which bacterial species percentage represents the mature dual-species biofilm. To reduce the impact of this limitation, homogenizing the dual-species biofilms and plating the bacterial suspensions in selective agar media allow the determination of the number of culturable cells of each microbial species, although this strategy is time-consuming and sometimes does not allow the online monitoring of antibiotic susceptibility testing of a high number of samples or not in a high-throughput screening.

To the best of our knowledge, a reproducible, fast, easy-to-perform, and reliable method for the measurement of dual-species bacterial biofilms is greatly desired. New strategies that allow the study of dual-species biofilms in all their complexity need to be developed. The present study describes an applicable methodology based on the use of bacteria constitutively expressing fluorescent or bioluminescent proteins, and as a proof of concept, we specifically used the strains of *Pseudomonas aeruginosa* and *Burkholderia cenocepacia* species. These microbes have remarkable clinical relevance producing severe nosocomial infections associated with the formation of polymicrobial biofilms, especially in cystic fibrosis and other immunocompromised patients [22]. The formation of dual-species biofilms with these strains in microtiter plates using this developed methodology allows their independent quantification, making it possible to determine the percentage of each bacterial species within the biofilm in a reproducible, fast, and reliable way and without the necessity of using advanced microscopy techniques.

## 2. Materials and Methods

### 2.1. Bacterial Strains and Culture Conditions

Reference laboratory strains of *P. aeruginosa* PAO1 (ATCC 15692) and *B. cenocepacia* J2315 (ATCC BAA-245) were grown in Luria–Bertani (LB; tryptone 10 g/L, yeast extract 5 g/L, and sodium chloride 10 g/L) medium (Scharlab, Barcelona, Spain) and in tryptic soy broth (TSB; casein peptone 17 g/L, soy peptone 3 g/L, sodium chloride 5 g/L, dipotassium phosphate 2.5 g/L, and dextrose 2.5 g/L) medium (Scharlab, Barcelona, Spain) at 37 °C and 30 °C, respectively. *P. aeruginosa* PAO1::eGFP (MK171) [23] constitutively expressing a green fluorescent protein marker from the chromosome (eGFPmut3) and *P. aeruginosa* PAO1 chromosomally modified by the integration of a mini-Tn7-*lux* constitutively expressing the luciferase genes [24] were also used, supplementing the media with 50 μg/mL gentamicin as the selective pressure. *B. cenocepacia* was transformed with pETS248-Tc-E2Crimson plasmid (see the plasmid construction and bacterial transformation and its low copy number vector) and growth supplementing media with 80 μg/mL of tetracycline. The strains were preserved in 20% glycerol stocks at −80 °C and were freshly revived for each experiment.

### 2.2. Plasmid Construction and B. cenocepacia Transformation

We constructed a plasmid expressing an E2-Crimson red fluorescent protein. Recombinant DNA techniques were performed using standard procedures on *Escherichia coli* DH5α cells [25]. To construct this plasmid, the promoter probe vector pETS134-GFP containing the promoter region of the *P. aeruginosa nrdA* gene was used as the backbone [26]. The gentamycin resistance gene fragment was removed from pETS134-GFP by double *BglII* and *NsiI* digestion (Thermo Fisher, Waltham, MA, USA). In parallel, a tetracycline resistance gene was amplified from the pEX18Tc vector [27] by a polymerase chain reaction (PCR) using the primers TcR_Fw (5′-GCGATGCATGACGTCAGGTGGCACTTTTC-3′; *NsiI*) and TcR_Rv (5′-GCGAGATCTATTCACAGTTCTCCGCAAG-3′; *BglII*). The resulting DNA amplification was gel-purified using a GeneJET Gel Extraction Kit (Thermo Fisher, Waltham, MA, USA), digested with *NsiI* and *BglII*, and ligated into the pETS134-GFP band fragment without gentamycin, generating the pETS247-Tc-GFP plasmid. Next, the GFP present in this plasmid was replaced by the E2-Crimson red fluorescent protein by digesting pETS247-Tc-GFP with *ApaI* and *NdeI*. Then, the DNA encoding the E2-Crimson fluorescent protein was amplified from the pUCP20T-E2Crimson vector [28] by PCR using the primers E2C_Fw (5′-GGGCATATGGGCGAGCTCGATAG-3′; *NdeI*) and E2C_Rv (5′-ATTGGGCCCTTACAATTCGTCGTGCTTGTAC-3′; *ApaI*). The amplified DNA fragment corresponding to E2-Crimson was gel-purified using a GeneJET Gel Extraction Kit (Thermo Fisher, Waltham, MA, USA), digested with *ApaI* and *NdeI* (Thermo Fisher, Waltham, MA, USA), and ligated into the digested pETS247-Tc-GFP to form pETS248-Tc-E2Crimson. The amplified DNA fragment corresponding to E2-Crimson was gel-purified using a GeneJET Gel Extraction Kit (Thermo Fisher, Waltham, MA, USA), digested with *ApaI* and *NdeI* (Thermo Fisher, Waltham, MA, USA), and ligated into the digested pETS247-Tc-GFP to form pETS248-Tc-E2Crimson.

For *B. cenocepacia* transformation, a previously easily described procedure was used [29]. The electroporation parameters were 2.5 kV and 25 μF using a GenePulser Xcell Electroporation System (Bio-Rad, Hercules, CA, USA). The DNA-to-cell ratio was 0.45 μg plasmid DNA in 150 μL of competent bacterial cells.

### 2.3. Plasmid Maintenance Depending on Antibiotic Selection Pressure in the B. cenocepacia Transformed Strain

The *B. cenocepacia* transformed strain integrity was confirmed by images detecting E2-Crimson in ON culture cells under a Nikon inverted fluorescence microscope, ECLIPSE Ti-S/L100 (Nikon, Tokyo, Japan). To test the plasmid maintenance in the transformed *B. cenocepacia* strain J2315 carrying pETS248-Tc-E2Crimson, we developed static 48 h biofilms in microtiter plates with or without the selective antibiotic (80 mg/mL of tetracycline). After 48 h, the biofilm was resuspended with 100 μL of 1× phosphate-buffered saline (PBS) (Fisher Scientific, Hampton, VA, USA), and the absorbance (Abs_550nm_) and relative fluorescence units (RFU) of E2-Crimson per well (excitation at 580 ± 20 nm and emission at 635 ± 35 nm) were measured in a Spark multimode microplate reader (Tecan, Männedorf, Switzerland).

### 2.4. Development of a Microtiter Plate Screen for Dual-Species Biofilm Biomass Quantification

In this work, as a proof of concept, we established a dual-species biofilm of a luminescence-expressing strain of *P. aeruginosa* and a fluorescence-expressing strain of *B. cenocepacia*. Overnight cultures of *P. aeruginosa* PAO1 chromosomally carrying the mini-Tn7-*lux* [24] and *B. cenocepacia* J2315 transformed with pETS247-Tc-E2Crimson were grown with the corresponding antibiotics in LB and TSB medium (Scharlab, Barcelona, Spain), respectively. Note that, in the case of undertaking studies using other bacterial species, different vectors with replication compatibility should be used. Then, both ON cultures were adjusted to OD_550nm_ = 0.1, corresponding to 10^7^ colony-forming units (CFU/mL), in TSB + 0.2% glucose without antibiotic pressure (Panreac Quimica, Barcelona, Spain) using a UV Mini 1240 spectrophotometer (Shimadzu, Kyoto, Japan) (see Figure 1A). The use of glucose is necessary to promote the development of biofilm in these species [30,31]. Then, 200 µL/well was added to Costar^®^ 96-well flat black polystyrene plates (Corning, NY, USA), and the plates were incubated statically at 37 °C in a compact mini-incubator (Labnet International, Edison, NJ, USA) with humidity saturation conditions to allow cell adhesion to the plastic walls and biofilm formation in the air–liquid interface. After 48 h, planktonic bacteria were removed by washing three times with 300 µL/well of 1× PBS (pH 7.4) to retain exclusively the attached biofilm-forming bacteria. At this point, for the development of the dual-species biofilm, strains prepared from freshly ON culture were inoculated together into wells containing a previously formed monomicrobial biofilm following the same steps previously explained (a schematic representation is shown in Figure 1). After an additional 24 h of incubation, the wells were washed 3× with PBS to remove planktonic bacteria, and the cell biomass of the biofilms was quantified using crystal violet staining. Fluorescence, bioluminescence, or CFU counting was carried out. Wells filled with the medium alone were used as a negative control, and the average values of fluorescence and bioluminescence reported in these wells were subtracted from all the data.

For the crystal violet staining procedure (see Figure 1B), 200 μL/well methanol (Fisher Scientific, Hampton, VA, USA) was added for 15 min to fix the biofilms. Then, the methanol was removed, and the wells were allowed to evaporate until they were completely dry. Then, 200 μL/well of crystal violet (1%) (Sigma-Aldrich, St. Louis, MO, USA) was added for 5 min, and the excess dye was removed through water washes. Finally, the crystal violet was dissolved with 200 μL/well of acetic acid (33%) (Fisher Scientific, Hampton, VA, USA) under agitation (approximately 150 rpm), and the absorbance was measured at 570 nm for the semiquantitative analysis of the biofilm biomass. The fluorescence and bioluminescence of the formed biofilms were quantified simply by homogenizing the dual-species biofilms in PBS through strong pipetting and pipette tip scraping on the walls of the wells (see Figure 1B).

The values of crystal violet absorbance (Abs_570nm_), lux luminescence (relative light units, RLU), and E2-Crimson fluorescence expression (excitation at 580 ± 20 nm and emission at 635 ± 35 nm) were quantified independently in each well using a Spark^®^ multimode microplate reader (TECAN). To establish the percentages of growth of each species in the dual-species biofilms, the level of fluorescence/bioluminescence of monospecies biofilms grown under the same conditions in the same microtiter plate and during the same time was quantified, and the mean of these values was established as 100% biofilm growth.

### 2.5. Colony-Forming Unit Counts from Biofilms

To quantify the biofilms by CFU counts, the bacterial biofilm formed on the walls of each well was removed by scraping with a pipette tip and resuspended by pipetting in 100 μL of 1× PBS. Then, each biofilm cell suspension was transferred to a 1.5 mL Eppendorf tube, disaggregated by pipetting, and vortexed for 30 s to separate the cells in the biofilm. Bacterial suspensions were then serially diluted in 1× PBS and plated on selective tryptic soy agar (TSA; casein peptone 15 g/L, soy peptone 5 g/L, sodium chloride 5 g/L, and agar 15 g/L) medium (Scharlab, Barcelona, Spain). TSA plates and TSA plates supplemented with 2 μg/mL ciprofloxacin (Sigma-Aldrich, St. Louis, MO, USA) were used to quantify *P. aeruginosa* and *B. cenocepacia*, respectively. After 24 h and 48 h of incubation of *P. aeruginosa* and *B. cenocepacia*, respectively, the colonies were counted, and the CFU/mL were determined. Ciprofloxacin was used since it inhibits the growth of *P. aeruginosa* but allows the growth of *B. cenocepacia*. On TSA plates, *P. aeruginosa* grows earlier due to its higher replication speed, preventing the growth of *B. cenocepacia*, while, on TSA plates supplemented with ciprofloxacin, only *B. cenocepacia* grows.

### 2.6. Fluorescence Microscopy to Visualize Biofilm Formation

Monomicrobial or dual-species biofilms were developed in microtiter plates, as described above. Briefly, 200 µL/well of *P. aeruginosa* or *B. cenocepacia* ON suspensions with an OD_550 nm_ = 0.1 were inoculated on the microtiter plates and incubated in TSB + 0.2% glucose at 37 °C, maintaining the humidity saturation conditions over 48 h. After this time, the wells underwent three rounds of washing with PBS to remove planktonic bacteria. Dual-species biofilms were performed by the inoculation and growth of *P. aeruginosa* or *B. cenocepacia* suspensions into wells that already contained a developed monomicrobial biofilm.

For microscopic observation, biofilms developed at the air–liquid interface were carefully detached from the walls of the wells using a micropipette tip, trying to minimize the possible decomposition or alteration of the original biofilm structure, and placed on a microscope slide (Thermo Fisher, Waltham, MA, USA). The appearance of the biofilm was analyzed with a Nikon inverted fluorescence microscope, ECLIPSE Ti-S/L100 (Nikon, Tokyo, Japan), coupled with a DS-Qi2 Nikon camera (Nikon, Tokyo, Japan). Fluorescent representative images were obtained with a 100× objective and subsequently processed using Fiji ImageJ software [32].

## 3. Results and Discussion

To establish a high-throughput system to precisely quantify the relative abundance of each microbial species in a dual-species biofilm, bacteria were labeled with designed low copy number vectors to express proteins that emit fluorescence or bioluminescence or chromosomically inserted in a single copy. In this study, we worked with *P. aeruginosa* PAO1 with a mini-Tn7-*lux* chromosomically inserted, a previously described strain constitutively expressing bioluminescence [24], and a *B. cenocepacia* J2315 strain transformed with pETS248-Tc-E2Crimson, a plasmid that expresses a red fluorescent protein (see Section 2.2). However, any plasmid with these properties can be used to label any biofilm-forming bacteria of interest.

These two species were selected due to their frequent simultaneous presence in polymicrobial or dual-species biofilms related to severe respiratory airway nosocomial infections. A critical clinical trait of *P. aeruginosa* is its capacity to interact and coexist with other microorganisms in multispecies communities [33,34]. For example, bronchiectasis produced by chronic obstructive lung disease (COPD), cystic fibrosis (CF), and lung injury, among others, is usually related to *P. aeruginosa* and *B. cenocepacia* coinfections. Furthermore, these species demonstrate a propensity to associate intimately with polymicrobial biofilm communities [35], so its study in dual-species biofilms has key biological relevance.

The strains of *P. aeruginosa* and *B. cenocepacia* used in this study contain resistance genes against gentamicin and tetracycline as selection markers, respectively. However, the expression of luciferase (lux) and E2-Crimson was not affected independently of the presence/absence of an antibiotic. *P. aeruginosa* mini-Tn7-*lux* was chromosomically inserted in a single copy expressing itself constitutively, and in *B. cenocepacia*, the plasmid was maintained in the cytoplasm, and the expression of E2-Crimson was maintained without an antibiotic when the strain developed in a static biofilm (Figure 2). Our strains were free of different antibiotic pressures during the development of the experiment.

Once it was demonstrated that the antibiotic selection pressure was unnecessary, the biofilm development experiments were carried out without antibiotics in the culture medium. We first validated our method by growing monomicrobial biofilms of *P. aeruginosa* PAO1 mini-Tn7-*lux* and *B. cenocepacia* pETS248-Tc-E2Crimson and quantifying their biomass by directly measuring the crystal violet and their fluorescence and bioluminescence (Figure 1) (see Section 2.4). Once the biofilm was formed, the reproducibility of both strategies was further evaluated, showing that the coefficient of variation (CoV; the ratio of the standard deviation to the mean) of all biofilm biomass values was similar between biofilms quantified by both methods (crystal violet and fluorescence and bioluminescence) (Figure 3A). In the *P. aeruginosa* PAO1 microtiter plate biofilm, a 15% CoV was found between crystal violet replicates, while a slight decrease (11%) was observed using bioluminescence measurements. The same tendency was observed in *B. cenocepacia* biofilms but with the highest variability, with 47% CoV in the crystal violet versus 31% in the fluorescence measurements (Figure 3A). The variability of the biofilm biomass quantification in the *B. cenocepacia* biofilms between the different wells was higher than for the *P. aeruginosa* biofilms, which were more homogeneous between replicates, independently of the method of quantification used. These results confirmed that using bacterial label tags allows biofilm quantification results comparable to those obtained with the traditional crystal violet, avoiding its limitations and reducing the variability between replicates. Once the validity of the measurements was demonstrated, dual-species biofilms of *P. aeruginosa* and *B. cenocepacia* were established to quantify each biofilm-forming species independently (Figure 3B). Note that using crystal violet will quantify the total biomass without distinguishing the specific contribution of each bacterial species. 

Monomicrobial mature biofilms of *P. aeruginosa* PAO1 mini-Tn7-*lux* and *B. cenocepacia* pETS248-Tc-E2Crimson were first established (48 h) and further co-inoculated with a culture of the opposite species for an additional 24 h to characterize their effect on the original mature biofilm (Figure 1). The results showed that the addition of *P. aeruginosa* significantly reduced the initial *B. cenocepacia* biofilm by approximately 50% (Figure 3B). Detailed observation by fluorescence microscopy showed that *B. cenocepacia* remained in the base of the biofilm well, and the growth of *P. aeruginosa* began to infringe on it (Figure 4). Microscopic images confirmed the coexistence of both bacterial species in biofilm growth. Previous dual-species biofilm studies have already reported a decrease in *B. cenocepacia* cells when *P. aeruginosa* was present [35]. However, no direct growth-inhibitory activity has been proven between the two species. It was also seen that, in 24 h, *P. aeruginosa* was accountable for 80% of the biofilm cell abundance in a mature biofilm (72 h), despite growing together with *B. cenocepacia*. This result demonstrates the expected dominant profile of *P. aeruginosa* over *B. cenocepacia,* as reported in previous studies [33,36], even early in cocultures, validating our method.

On the other hand, the growth of the *P. aeruginosa* biofilm remained unchanged when *B. cenocepacia* was later inoculated over a mature biofilm (48 h), reaching similar values to those of the monomicrobial *P. aeruginosa* biofilm (100% growth). However, *B. cenocepacia* grew and settled into the formed *P. aeruginosa* biofilm, although its growth was only 35% of the total growth of a monomicrobial *B. cenocepacia* mature biofilm (Figure 3B). These results obtained through quantification by fluorescence and bioluminescence measurements were verified by observing the appearance of the biofilms by fluorescence microscopy. While *B. cenocepacia* forms a dense biofilm when it grows alone, its development is seriously compromised in the coculture with *P. aeruginosa*, since the latter is the predominant species (Figure 4). Importantly, we observed the establishment of a *B. cenocepacia* biofilm over a mature *P. aeruginosa* biofilm, which always showed some dominant growth over other types of bacterial species [33].

The results obtained by the fluorescence and bioluminescence measurements were further validated by analyzing the CFU counts for every tested condition. As Figure 3C shows, the CFUs of *P. aeruginosa* were stable regardless of the inoculation time point and the presence of *B. cenocepacia* in the culture. In contrast, the CFUs/mL of *B. cenocepacia* were the highest in the absence of *P. aeruginosa,* whereas there was a significant decrease in CFUs after the addition of *P. aeruginosa* to a mature *B. cenocepacia* biofilm. Specifically, the number of *B. cenocepacia* colonies obtained in the dual-species biofilms was three orders of magnitude lower than that in the control monomicrobial biofilms. When incubated for 24 h over a *P. aeruginosa* mature biofilm, the viability of *B. cenocepacia* decreased compared to its monoculture, although it was still able to form a biofilm. These results are in line with the fluorescence and bioluminescence measurements described in Figure 3A,B and therefore validate our biofilm quantification method.

Using genetically transformed bacterial strains to express fluorescence and bioluminescence in microtiter plates has excellent advantages in studying dual-species biofilms. In this study, a broad host range of replicative plasmids with the origins of the replications compatible for the tested bacteria was used. However, using recombinant strains containing antibiotic resistance genes could limit the assays focused on antibiotics. Therefore, the choice of the selection marker in the vector to avoid interference in the study is crucial. Different constructions and plasmids will be needed if other microorganisms are used in further studies. 

The most important advantage of this method is the possibility of performing individualized quantifications of each bacterial species forming a dual-species biofilm, determining the percentage that each one occupies. It facilitates the independent monitoring of each species in dual-species biofilm evolution and the measurement of solely live microbial cells by preventing the overestimation of the biomass by measuring dead cells in the biofilm matrix. In this sense, the constitutive expression of the fluorescence/bioluminescence proteins shown in our method could be stopped in dormant cells in biofilms and may produce an underestimation of a biofilm biomass being a methodological noncontrollable limitation. Note that, in most of the current dual-species biofilm biomass quantification methods for static setups, it is impossible to easily evaluate the contribution of each bacterial species to biofilm development [37,38]. Additionally, this problem is accentuated during the evaluation of different antimicrobial therapies and the study of the specific effect of each bacterial species on a dual-species biofilm [39]. 

Furthermore, the biofilm quantifications reported here by fluorescent and bioluminescent strains are reproducible, with the coefficients of variation between replicates lower than those reported using the traditional crystal violet method, reflecting improved reproducibility. Also, the fluorescent and bioluminescent procedure is faster and more direct than that of crystal violet, which involves several steps with waiting periods. In addition, fluorescence is a methodology broadly used to evaluate and quantify biofilm formation [2,23].

## 4. Conclusions

The study and characterization of dual or polymicrobial static biofilms in vitro remain challenging, and the most widely employed method for this purpose is the development of biofilms in microtiter plates. However, the lack of specificity of the current methods hinders the individual identification and quantification of each species within a biofilm. In this context, this study demonstrated that utilizing genetically modified strains emitting fluorescence or bioluminescence enables the rapid, reproducible, and reliable determination of individual species proportions within a biofilm. These biofilm quantification results, as well as the reliability of the genetically transformed strains, have been verified and validated using time-demanding techniques such as CFU counting and observation through fluorescence microscopy. 

Despite its significant advantages, the application of the presented method requires attention to specific details. It is essential to genetically transform the desired strains and to carefully select the appropriate selectable marker (often an antibiotic resistance gene) within the vectors to avoid interference with subsequent studies. In essence, we presented a straightforward, faster, and more reproducible strategy compared to the current methods for quantifying dual-species biofilms.

## Figures and Tables

**Figure 1 microorganisms-11-02244-f001:**
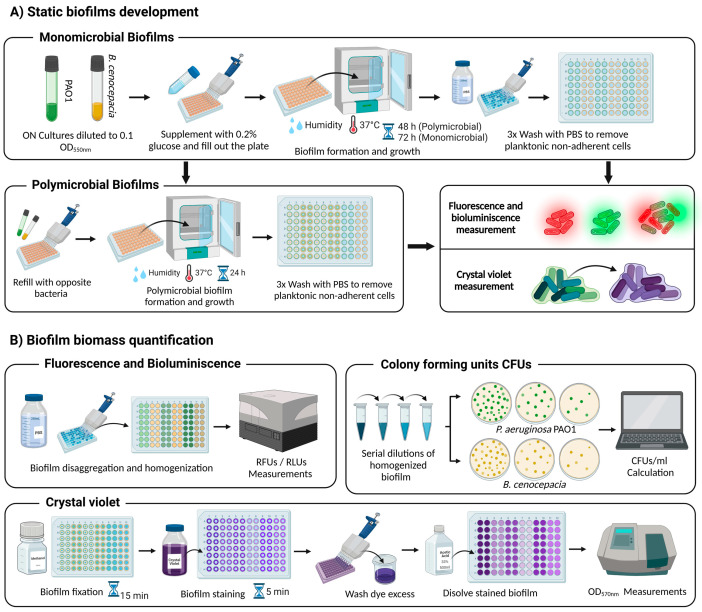
Schematic summary of the microtiter plate assay for the formation and quantification of static biofilms. (**A**) The procedures followed to form monomicrobial and dual-species biofilms. (**B**) Quantification methods used for the analysis of biofilms: CFU counting, crystal violet staining, and fluorescence (RFU, relative fluorescence units) or bioluminescence (RLU, relative light units) measurements developed as a novel, reproducible, and reliable alternative method.

**Figure 2 microorganisms-11-02244-f002:**
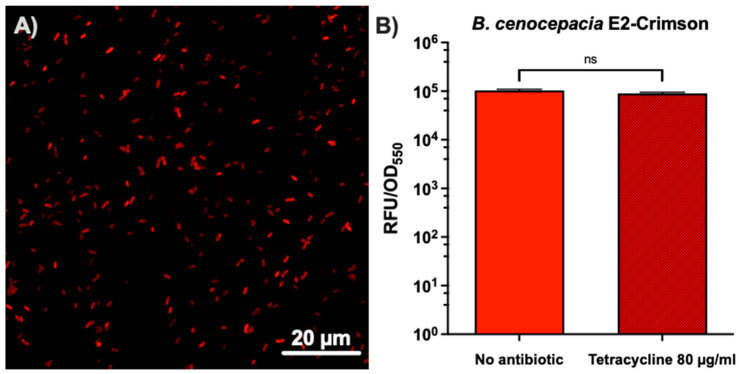
Evaluation of E2-Crimson fluorescence expression in the *B. cenocepacia* transformed strain, depending on the presence/absence of the plasmid selection antibiotic tetracycline. (**A**) Fluorescence microscopy image at 100× of an ON culture from a successfully electrotransformed *B. cenocepacia* J2315 expressing the E2-Crimson fluorescent protein. (**B**) Relative fluorescent units (RFU)/OD_550nm_ comparison of three independent replicates (n = 12) after 48 h of biofilm-forming *B. cenocepacia* E2-Crimson growth with and without the presence of 80 μg/mL of tetracycline. ns, nonsignificant differences (unpaired *t*-test).

**Figure 3 microorganisms-11-02244-f003:**
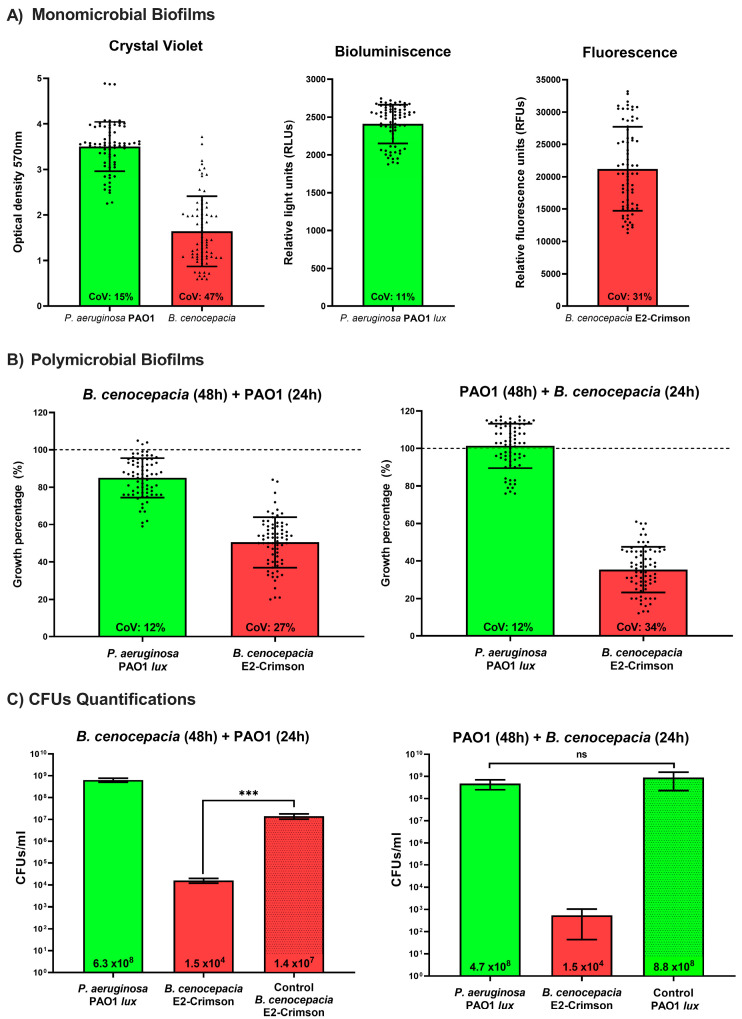
Assays validating the novel biofilm biomass quantification method for measuring fluorescence and bioluminescence. (**A**) Reproducibility test comparing the crystal violet assay with the bioluminescence and fluorescence assay methodology on monomicrobial biofilms; the results correspond to three independent replicates (n = 72). (**B**) Bioluminescence quantification of *P. aeruginosa* lux and *B. cenocepacia* E2-Crimson forming dual-species biofilms. The 100% biofilm growth value is the average expression reported by a monomicrobial biofilm grown for 72 h under the same conditions as the dual-species biofilms. The values correspond to three independent experiments (n = 72); the data are presented as the mean ± standard deviation. The coefficient of variation (CoV) shows the variability in relation to the mean in each condition. (**C**) Colony-forming unit (CFU) quantification of *P. aeruginosa* lux and *B. cenocepacia* E2-Crimson forming mono- and dual-species biofilms; the results correspond to four CFU recounts of each condition. *** *p* < 0.001; ns, nonsignificant differences (unpaired *t*-test).

**Figure 4 microorganisms-11-02244-f004:**
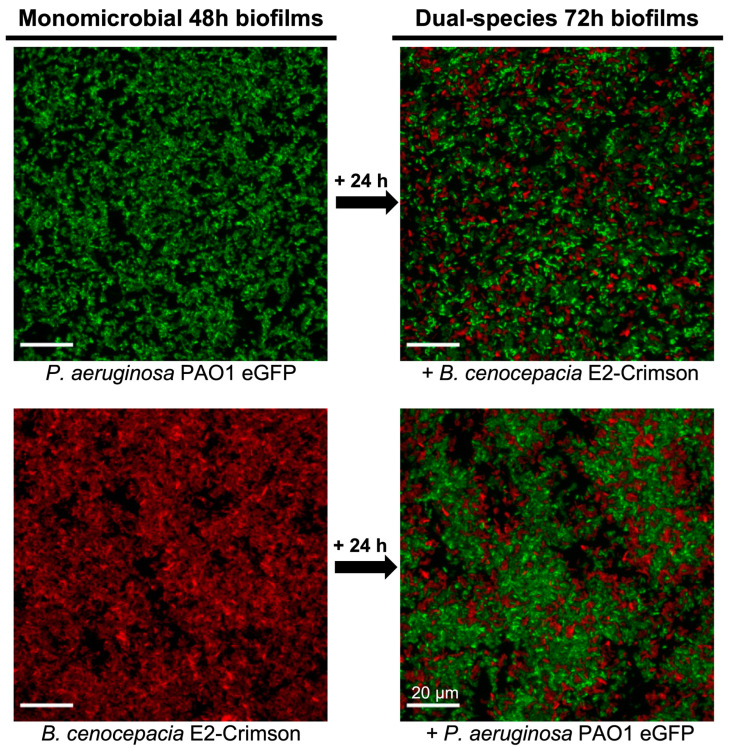
Representative fluorescence microscopy images of *P. aeruginosa* and *B. cenocepacia* monomicrobial and dual-species biofilms. *P. aeruginosa* eGFP is reported in green, and *B. cenocepacia* E2-Crimson is reported in red. Monomicrobial biofilm wells were grown for 48 h and then subsequently inoculated with the opposite bacteria to form dual-species biofilms (growth for an additional 24 h).

## Data Availability

Research data will be shared by the corresponding authors upon request.

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
