# Peer review of "A High-Throughput Microtiter Plate Screening Assay to Quantify and Differentiate Species in Dual-Species Biofilms"

_microorganisms, 2023, doi:10.3390/microorganisms11092244_

Round 1

Reviewer 1 Report

Comments to the authors:

Line 28: Please write full name of strains in the Keywords.

Line 100: Change Burkholderia cenocepacia to B. cenocepacia

Please provide the composition of LB and TSB or add a reference.

Please provide the full name of PBS, CFU, RFU, and RLU in their first appearance.

Line 198: keep only the abbreviation CFU and write the full name in line 154.

Please provide TSA composition or add a reference

2.6 Section: Should be rewritten. Add how cultures were grown, in which medium, temperature, volume, duration, after how authors obtain biofilm (centrifugation?), etc. Did the authors detect the chemical composition of biofilms or only by microscopy?

The conclusion section is absent. It could be better to add a 2-3 paragraph summarizing the main and important findings which will be easy to follow for outers. 

What about the other strains? Will this method easy could be provided for others? What are the disadvantages and advantages of high-throughput microtiter plate screening?

Reference: More recent references should be added.

Author Response

Enclosed file.

Reviewer 2 Report

The manuscript by Campo-Perez et al. described novel quantitative strategies to quantify and differentiate species in dual-species biofilm, for strains that can constitutively express fluorescent or bioluminescent proteins.

The manuscript needs minor revisions to reach the quality required for publication in the journal.

-        INTRODUCTION: One of the most serious challenges to researchers is the emergence of multi-drug resistance pathogenic bacteria that form biofilm and rapidly develop resistance to currently available antibiotics for the treatment of human infections. I suggest emphasizing the importance of biofilm formation as a valuable target to fight against severe chronic infections. I suggest reading and adding the recent references:

-      Life (Basel). 2023 Jan 6;13(1):172. doi: 10.3390/life13010172.

-      Int J Pharm. 2023 Jan 25;631:122492. doi: 10.1016/j.ijpharm.2022.122492.

-      Curr Med Chem. 2022;29(25):4307-4310. doi: 10.2174/0929867329666220103095551

-      Int J Mol Sci. 2023 Mar 2;24(5):4872. doi: 10.3390/ijms24054872.

-        The conclusion paragraph in the manuscript is missing. Please add.

-        The experiment should be repeated multiple times before it is reported. It is not clear how many times each experiment was performed.

Minor editing of English language required

Author Response

Enclosed file

Round 2

Reviewer 1 Report

Dear Authors, 

Thank you for considering the reviewer's comments and significantly improving the manuscript.